# Contextual Factors Associated with Abuse of Home-Dwelling Persons with Dementia: A Cross-Sectional Exploratory Study of Informal Caregivers

**DOI:** 10.3390/ijerph20042823

**Published:** 2023-02-05

**Authors:** Gunn Steinsheim, Wenche Malmedal, Turid Follestad, Bonnie Olsen, Susan Saga

**Affiliations:** 1Department of Public Health and Nursing, Norwegian University of Science and Technology (NTNU), 7030 Trondheim, Norway; 2Department of Clinical and Molecular Medicine, Norwegian University of Science and Technology (NTNU), 7030 Trondheim, Norway; 3Clinical Research Unit Central Norway, St. Olavs Hospital, 7030 Trondheim, Norway; 4Department of Family Medicine, Keck School of Medicine of the University of Southern California, Los Angeles, CA 91803, USA

**Keywords:** elder abuse, dementia, Norway, survey, family caregivers, lasso, penalized regression

## Abstract

One in six community-dwelling older adults experience elder abuse yearly, and persons with dementia are especially at risk. Although many risk factors for elder abuse have been identified, there are still knowledge gaps concerning risk and protective factors. This cross-sectional survey among Norwegian informal caregivers (ICGs) aimed to find individual, relational, and community factors associated with psychological and physical abuse of home-dwelling persons with dementia. This study involved 540 ICGs and was conducted from May to December 2021. Statistical analysis using penalized logistic regression with lasso was performed to find covariates associated with psychological and physical elder abuse. The most prominent risk factor for both subtypes of abuse was the caregiver being a spouse. In addition, risk factors for psychological abuse were having a higher caregiver burden, experiencing psychological aggression from the person with dementia, and the person with dementia being followed up by their general practitioner. For physical abuse, the protective factors were the ICG being female and having an assigned personal municipal health service contact, while the risk factors were the ICG attending a caregiver training program and experiencing physical aggression from the person with dementia and the person with dementia having a higher degree of disability. These findings add to the existing knowledge of risk and protective factors in elder abuse among home-dwelling persons with dementia. This study provides relevant knowledge for healthcare personnel working with persons with dementia and their caregivers and for the development of interventions to prevent elder abuse.

## 1. Introduction

Elder abuse can be defined as “a single or repeated act, or lack of appropriate action, occurring within any relationship where there is an expectation of trust which causes harm or distress to an older person” [1]. It can take many forms, but five subtypes are often mentioned: psychological, physical, sexual, and financial abuse and neglect. Elder abuse may adversely affect the mental and physical health of the victim, and premature mortality and depression are the best-documented outcomes [2]. With one in six community-dwelling older adults experiencing elder abuse yearly [3], it threatens the health, well-being, and human rights of older persons globally. Considering the gradually aging population worldwide, abuse is a growing problem. Persons with dementia are especially at risk, and WHO estimates that as many as two out of three persons with dementia experience some form of abuse [4]. The prevalence of dementia is estimated to increase as the population ages. In Norway, more than 100,000 persons are living with dementia, and it is estimated that the number will more than double in the next 30 years [5]. Therefore, it is crucial to increase the knowledge of mechanisms affecting elder abuse, especially among persons with dementia, to raise awareness and optimize the prevention and detection of this pressing public health issue.

Elder abuse is a complex and multifaceted phenomenon, and although various interventions to prevent or reduce the occurrence of abuse have been developed, the evidence for the effectiveness of the interventions is sparse [6]. To address the complexities of this phenomenon, Roberto and Teaster [7] proposed the Contextual Theory of Elder Abuse. The theory consists of four contexts—individual, relational, community, and societal. An outline of the framework with examples of related constructs/factors is shown in Figure 1. Individual characteristics, relationships with others, offers and responses in the community, and societal norms, values, and policy all intertwine and affect the occurrence of and responses to elder abuse. In studies investigating the abuse of home-dwelling persons with dementia, a wide variation in potential risk factors has been identified. The most commonly identified individual risk factors in relation to the person with dementia include high levels of behavioral and psychological symptoms of dementia (BPSD) [8,9,10,11], aggression towards the caregiver [10,12,13], more severe cognitive impairment [8,9,13], and low physical function [13,14]. Factors related to the informal caregiver (ICG) are psychological symptoms related to depression [10,12,13,15] and anxiety [10,15]. Relational risk factors are high levels of caregiver burden or stress [8,10,12,16], poor or conflicting relationships [11], and more co-residing days [11,16]. At the community level, low social support for the caregiver [10,14] and less use of formal support [15,16] are identified as risk factors. In addition, societal factors, such as cultural norms and values, are potential risk factors [17], and ageism is one of the major societal risk factors [18].

Although several previous studies have explored this topic, knowledge gaps remain concerning risk and protective factors. In particular, studies exploring factors related to the community are underrepresented [19]. One of the five WHO priorities to tackle elder abuse in the next decade involves an increased understanding of protective factors in general and risk factors within a community and societal context [18]. To increase knowledge of the risk and protective factors of abuse of older persons with dementia, the contextual theory of elder abuse by Roberto and Teaster [7] guides the development of this study’s research questions and measurements. Using self-reported data from a convenience sample of ICGs across Norway and a statistical method for variable selection, this study aimed to find individual, relational, and community factors associated with psychological and physical abuse of home-dwelling persons with dementia.

## 2. Materials and Methods

This is a cross-sectional, anonymous, self-administered pen-and-paper survey and is part of a more extensive study exploring aspects of caregiver burden and elder abuse among ICGs of home-dwelling persons with dementia in Norway [20,21]. This study was conducted following the STROBE guidelines for cross-sectional studies [22].

### 2.1. Setting

This study was conducted among ICGs of home-dwelling persons with dementia in Norway. Although it was not the intention, the data collection took place in 2021 during the coronavirus pandemic. National restrictions to reduce the outbreak of COVID-19 were first implemented in March 2020 and continued with various measures throughout 2020 and 2021. National restrictions varied during 2021, with stricter regulations introduced in March and a gradual reduction until the end of September 2021 [23]. In addition, local restrictions varied due to local outbreaks. Rules regarding social distancing affected services for persons with dementia, especially daycare activities that functioned as social meeting arenas. Results from studies in the first months of the lockdown showed an increase in informal care among co-residing ICGs [24] and a worsening of BPSD among persons with dementia [25]. It is possible that these impacts were still present during data collection. 

In Norway, almost 70% of persons with dementia live at home [26]. Of these, 90% receive informal care from a relative or acquaintance, around half receive home nursing services, and approximately 20% visit a daycare center once or twice a week [26]. Primary care, such as home nursing services and homecare services, daycare activities, and dementia education, is the responsibility of the municipalities [27]. In addition to these public services, volunteer organizations offer cultural and social activities [28].

### 2.2. Participants and Data Collection 

The inclusion criteria were as follows: (1) the ICGs were adult family members or acquaintances of the person with dementia; (2) the ICGs had personal contact with the person with dementia at least once a week; (3) the person with dementia resided at home. Where there was more than one ICG, respondents were preferably the individual providing the most care.

A convenience sample was obtained through three recruitment strategies (see Figure 2). The first involved mailing questionnaires directly to ICGs registered in the Norwegian Registry of Persons Assessed for Cognitive Symptoms (NorCog). The second was through collaboration with municipal healthcare services and the Norwegian Health Association’s local dementia associations from all regions of Norway, where contact persons in the organizations distributed questionnaires to eligible ICGs. The third involved informing ICGs of the project through social media and partner organizations, leading to self-enrollment. All participants received an envelope containing the questionnaire, a joint information letter and consent form describing the project to the ICG, an information letter adapted for the person with dementia, and a stamped return envelope. Informed consent was obtained by participants’ completion and submission of the questionnaire to ensure anonymity. 

### 2.3. Measurements

The questionnaire was developed in collaboration with several user organizations, cognitive interviews with former ICGs of persons with dementia, and a small pilot study to ensure face and content validity. A more detailed description of the questionnaire development can be found in a previous publication from the same study [21]. The results from the pilot study indicated a completion time of about 40 min for the complete questionnaire.

Abuse was measured by five items concerning psychological abuse and nine items concerning physical abuse. The items were adapted from two Norwegian cross-sectional studies of elder abuse conducted in nursing homes [29] and among home-dwelling older persons [30]. All items are shown in Figure 3. The ICGs were asked how many times during the past 12 months they had committed specific acts, with the response options “Never,” “Once,” “2–5 times,” “6–10 times,” and “More than 10 times.” The items of psychological and physical abuse were summarized to generate two bivariate outcome variables (“abuse” or “no abuse”), one for each subtype of abuse. The Pillemer criteria [31] were used to define abuse. For physical abuse, one or more episodes in the past 12 months were defined as abuse. For psychological abuse, at least 10 or more incidents in the past year were used. First, midpoints for the response categories were defined as follows: “Never” = 0, “Once” = 1, “2–5 times” = 3.5, “6–10 times” = 8, and “More than 10 times” = 12.5. The five psychological abuse items were then summarized. Adding midpoint values to create a sum score is consistent with previous studies [32,33]. To establish dichotomized scores, a sum score of 10 or more was categorized as abuse and less than 10 as no abuse.

Covariates were collected related to the individual, relational, and community contexts. A complete list of the measures is provided in Table 1. In addition, the ICGs reported their understanding of the type of dementia their care recipient was diagnosed with. 

### 2.4. Ethical Considerations

Due to the sensitive study topic and the expected discomfort of the participants, the questionnaire and study information was designed in collaboration with user organizations and ICGs. The written information emphasized that participation in this study was voluntary and anonymous. The ICGs were encouraged to inform the person with dementia about their participation in the project or to assess whether the person with dementia would have welcomed participation. This study was approved by the Regional Committee for Medical Research Ethics Central Norway (#153444).

### 2.5. Statistical Analysis

The statistical analyses were conducted using the Stata Statistical Software, Release 17 [41]. Continuous variables were summarized as mean values and standard deviations (SDs). Categorical variables were described using frequencies and proportions (percentages). 

To identify factors associated with psychological and physical abuse, penalized logistic regression using the least absolute shrinkage and selection operator (lasso) was used [42]. Lasso is a method that jointly performs variable selection and parameter estimation. The technique aims to identify the most important variables for outcome prediction, not to assess the degree of association for the individual variables. The regression coefficients are shrunk toward zero, reducing the chance of overfitting. The coefficients of variables that are only weakly associated with the outcome can be shrunk to exactly zero, leaving them out of the final model. The degree of shrinkage is determined by a penalty parameter, lambda (λ). The value of λ was determined by 10-fold cross validation, selecting the λ that results in the smallest estimate of the out-of-sample prediction error. Dummy variables for all levels of a categorical variable were included in lasso; hence, no level was set as a reference category. The linearity assumption of logistic regression for the continuous variables was tested by running univariable logistic regressions with continuous and categorized versions of the continuous variables and comparing the two models using the likelihood ratio (LR) test.

Lasso tends to select one of a set of strongly correlated covariates. Therefore, the association between all pairs of included variables was assessed to aid in interpreting the results. Spearman rank correlations or Cramer’s V was used to estimate the correlation between ordinal and/or continuous variables and between nominal variables, respectively. Further, Pearson’s chi-squared tests, Mann–Whitney U tests, or Kruskal–Wallis tests were used to assess the association between nominal variables, and between nominal and ordinal/continuous variables, as appropriate. The significance level was set to 0.05. To estimate the uncertainty of the lasso coefficients, the lasso procedure was repeated for 1000 bootstrap samples, and the proportion of bootstrap samples in which each variable was not set to zero was calculated. 

The level of missing data was acceptable. For the measures of psychological and physical abuse, 2.0% and 3.9% of the participants had at least one missing item, respectively. For the covariates, single-item measures had 0.2–3.0% missing values. In sum score scales, NPI-Q had the highest proportion of missing, with 8.0% of participants with at least one missing item. The rest of the sum score scales ranged from 2.2 to 3.9% of at least one missing item. Although the proportion of missing per variable was acceptable, it was randomly spread over the participants, such that 26.5% (143/540) of the participants had at least one missing value. Therefore, missing values for an item in sum score scales were replaced with the participants’ mean scale score of the observed items if a minimum of 75% of the items were answered. Little’s test for missing completely at random (MCAR) was performed on all items within each sum score scale with no significant results (*p* ranged from 0.08 to 0.99), indicating that missing items can be treated as MCAR. No other replacements were made for missing data. After imputation of sum score scales, the total number of participants with at least one missing value for the variables included in the multiple regression model was reduced to 18.7% (101/540). In the regression models, missing data are handled by using listwise deletion. 

## 3. Results

A total of 549 ICGs participated in this study from May to December 2021. After inspecting all the questionnaires, nine were excluded before analysis because the ICG did not meet the inclusion criteria. For details of the recruitment process, see Figure 2. The participating ICGs were aged 21 to 93 (mean 67.4, SD 11.8), 68.2% were female, and 42.4% were educated at a college or university level. The ICGs’ relation to the person with dementia was 63.1% spouses, and the rest were primarily children, plus relatives such as siblings, nieces and nephews, cousins, and grandchildren. The persons with dementia were aged 53 to 99 (mean 78.9, SD 7.8), and 48% were female. Furthermore, 43.9% were diagnosed with Alzheimer’s disease, and for 41.3%, the diagnosis was either mixed, unspecified, unknown, or the diagnostic process was still ongoing, while 14.9% had other specified types of dementia. Descriptive statistics by type of abuse are shown in Table 2. 

The number of variables selected by lasso was five for psychological abuse and nineteen for physical abuse (Table 3). Regarding psychological abuse, the estimated, penalized odds ratios indicate that a lack of experienced psychological aggression from the person with dementia is a protective factor (factors negatively associated with abuse), and being a spouse, having a higher caregiver burden, and the person with dementia being followed up by their general practitioner (GP) were risk factors (factors positively associated with abuse) (Table 3). These findings are supported by a relatively low uncertainty (80.5–100% not null, Table 3 and Figure 4A) and the distributions of the standardized coefficients (i.e., standardized coefficients on the log-odds scale) from the 1000 bootstrap samples, illustrated in Figure 5A. Helping the person with dementia every day was also selected as a risk factor by lasso in the original sample, but the results from the bootstrap samples indicate that the estimate is very uncertain (59% not null).

Regarding physical abuse, the estimated, penalized odds ratios indicate that the ICG being female, not experiencing physical aggression from the person with dementia, and having an assigned personal municipal health service contact were protective factors, while a higher degree of disability in the person with dementia, the ICG being a spouse, and attending a caregiver training program were risk factors (Table 3). These findings are supported by a relatively low uncertainty (83.5–99.7% not null) and by the distributions of the standardized coefficients in the bootstrap samples (Figure 5B). There was evidence that ICGs participating in caregiver training programs differed from those who did not participate on several covariates related to the burden of care. They had higher caregiver burden (*p* = 0.04, Mann–Whitney test), poorer mental health (*p* < 0.001, Mann–Whitney test), and a higher proportion provided daily care to the person with dementia (*p* = 0.02, chi-squared test) and were spouses of the person with dementia (*p* < 0.001, chi-squared test). Thirteen other covariates were selected by lasso (Table 3), but the results from the bootstrap samples indicate only weak evidence of an association with abuse (47.3–76.5% not null).

## 4. Discussion

This study aimed to identify factors associated with psychological and physical elder abuse among home-dwelling persons with dementia. The results indicated that the most important risk factors for psychological abuse were being a spouse, having a higher caregiver burden, experiencing psychological aggression from the person with dementia, and the person with dementia being followed up by their GP. For physical abuse, the ICG being female and having an assigned personal municipal health service contact were protective factors, while the ICG being a spouse, attending a caregiver training program, experiencing physical aggression from the person with dementia, and the person with dementia having a higher degree of disability were the most important risk factors.

### 4.1. Individual Context

The results indicated that female ICGs have a lower probability of committing physical abuse than males. Similar results were found in studies from the UK [43] and Japan [9], where male caregivers had a higher risk of overall abuse. In contrast, a study from Florida found that female caregivers had a higher risk of verbal abuse [12] but not physical abuse [13]. A potential explanation could be that men use physical restraint more frequently than women to prevent the person with dementia from being harmed, with males typically being physically stronger than females. A systematic review of risk factors in community-dwelling older persons found no consistent results regarding gender as a risk factor for either the perpetrator or the older person [19]. Hence, it is difficult to conclude whether gender is a risk factor or if this finding is due to other physiological, psychological, social, or cultural aspects related to sex and gender, which can affect the association with elder abuse.

In our study, the estimated probability of physical abuse increases with a higher degree of disability in persons with dementia, indicating disability is a risk factor for physical abuse. This result is consistent with findings from the US, where VandeWeerd, Paveza, Walsh, and Corvin [13] found high levels of functional impairment to be a risk factor for physical abuse. In contrast, studies from South Korea [44] and the UK [43] found reduced ADL functioning to be a protective factor against abuse. Although there is some inconsistency in the evidence, there is agreement that lower ADL functioning or frailty are risk factors for elder abuse in the general population [19,45,46]. We believe that disability increases the ICG’s burden due to the increased care needs of the person with dementia and the ICG’s concern for the individual’s health and well-being.

### 4.2. Relational Context

The ICG being a spouse is a potential risk factor for psychological and physical abuse. In their systematic review, Pillemer, Burnes, Riffin, and Lachs [46] found that it was more common for spouses or partners to be perpetrators of psychological and physical abuse in Europe, the US, and Israel, while children or children-in-law were the most common perpetrators in some Asian countries. This observation implies that cultural or societal factors affect this association. Additionally, the strong association between elder abuse and being a spouse might be confounded by the fact that most spouses co-reside with the person with dementia. Helping the person with dementia daily is also a potential risk factor for psychological and physical abuse, but with only weak or modest evidence in our data. This result might add to the speculation that co-residency is more critical than marital status. Co-residency has been identified as a risk factor for elder abuse in previous cross-sectional [16] and prospective [11] studies.

A higher ICG burden is associated with psychological abuse. Several studies involving persons with dementia have found associations between various types of elder abuse and caregiver burden [8,10,12]. Yan and Kwok [16] found a similar result with caregiver stress as a risk factor for verbal but not physical abuse inflicted by ICGs of older persons with dementia in Hong Kong. An ICG who feels stretched to the limit due to the burden of care may resort to verbal or psychological outbursts against the person with dementia, but it appears that an additional factor is needed for an ICG to become physically abusive. The results in Table 2 demonstrate that the burden is higher in the physical abuse group compared to the no physical abuse group. This finding implies an association between ICG burden and physical abuse, but the results of the lasso logistic regression show that other factors, such as disability of—and aggression from—the person with dementia, are stronger risk factors of physical abuse than caregiving burden. Another hypothesis is that verbal abuse may be more acceptable than physical abuse, both socially and juridically. This can increase the barrier against perpetrating physical abuse, but also increase the threshold for reporting physical abuse compared to psychological abuse. These social and legal barriers affect the underlying mechanisms of elder abuse.

ICGs not experiencing psychological aggression are estimated to have a lower probability of committing psychological abuse, while not experiencing physical aggression reduces the probability of committing physical abuse. Hence, psychological aggression by the person with dementia is a potential risk factor for psychological abuse. Correspondingly, physical aggression is a risk factor for physical abuse. ICGs experiencing aggression from the person with dementia have been identified as a risk factor for elder abuse in several previous studies [10,12,13]. These findings suggest bi-directionality in factors that lead to abuse. Longitudinal studies are needed to explore whether and to what extent ICGs resort to abusive behavior as a response to being abused themselves. Nevertheless, it is estimated that more than one in five ICGs experience severe aggression from the person with dementia [47], which can be traumatic for the ICG, especially if they themselves are in a vulnerable state. Therefore, it is crucial that healthcare personnel recognize aggressive behavior and assist ICGs in identifying prevention strategies.

### 4.3. Community Context

In the community context, having an assigned personal municipal health service contact is a protective factor against physical abuse. In previous qualitative research, ICGs of persons with dementia have reported that having a formal contact person is an important factor [48,49]. Community-based care coordinating interventions delivered by a professional may reduce BPSD and caregiver burden [50], and this is a possible explanation for the results in the present study. Future studies are needed to confirm causation and explore the effects of personal health service contacts on abuse risk.

In the present study, the person with dementia being followed up by their GP is a risk factor for psychological abuse. This was an unexpected finding, especially considering previous studies where formal care was seen to lower the risk of abuse [15,16,44]. It is possible that the association is caused by unidentified confounding factors. The reason the person with dementia contacts their GP might be another affliction, disease, or injury, which is the actual risk factor for psychological abuse.

Another unexpected finding is that ICGs attending a caregiver training program seem to have an increased risk of physical abuse. The purpose of the caregiver training program is to give ICGs increased knowledge of dementia and how to cope in a caregiver role. Similar interventions have positively affected ICGs and reduced caregiver burden [51], but have not significantly reduced abusive behavior [52]. In the present study, it is possible that ICGs who, for other reasons, have a higher risk of committing physical abuse seek caregiver training programs, with those other risk factors contributing to the training program appearing as a risk factor. This assumption is strengthened by the associations identified between attending a caregiver program and other covariates, such as caregiver burden and mental health. Even so, caregiver training would be expected to lower the risk of abuse. An alternative hypothesis is that caregivers who have attended a training program have gained more knowledge and awareness, thus reporting more abuse. This could be linked to the Dunning–Kruger Effect [53], which is a cognitive bias where people with low expertise or experience regarding a certain knowledge area tend to overestimate their ability and knowledge. Thus, ICGs with limited knowledge of dementia and caregiving might overestimate their ability in caregiving and not perceive their actions as potentially abusive.

### 4.4. Societal Context

Although the current study did not collect data from the societal context, cultural norms and values are potential risk factors for elder abuse and shape perceptions of caregiving and what constitutes elder abuse [54]. Ageism is an important factor [18]. While there is agreement that ageism is a risk factor for elder abuse, there is little research to confirm this causal association [55]. However, as described by Pillemer, Burnes, and Macneil [55], studies from other types of abuse, such as intimate partner violence and child maltreatment, have found that attitudes at the individual and societal levels influence the risk of abuse. Botngård et al. [56] found that healthcare personnel with poor attitudes toward people with dementia had a higher risk of committing abuse against care recipients. Although none of the above-mentioned studies directly examines the connection between ageism and elder abuse, they help form a theoretical basis for a causal connection. International studies are needed to further explore the association between elder abuse and societal factors such as legislation, norms, values, and culture.

### 4.5. Limitations

This study has limitations, so the results should be interpreted with caution. First, this is a cross-sectional study, so although we have used the terms risk and protective factors, causation cannot be established. This study was designed based on a contextual theory, and the results were discussed using relevant literature to strengthen assumptions about causality. However, much of the literature is also based on cross-sectional studies. Second, all the data were self-reported by ICGs and may therefore be affected by recall bias and under-reporting due to the stigma associated with elder abuse. We attempted to compensate for the latter by conducting this study anonymously. Third, there is a possibility that the COVID-19 pandemic and the various restrictions might have affected the results, especially the service provision covariates. Many restrictions were eased in 2021, but they varied across municipalities, and overall, there was an emphasis on measures that shielded frail and older persons from the risk of infection. Fourth, when considering the service provision covariates, simply measuring whether or not a service is used provides a limited contribution to understanding how it affects a complex outcome such as elder abuse.

## 5. Conclusions

While previous studies have tested a limited selection of potential risk factors, the present study used lasso on a large set of potential covariates to identify significant factors associated with the abuse of home-dwelling persons with dementia. We found risk and protective factors related to psychological and physical abuse in individual, relational, and community contexts, with the ICG being a spouse identified as the most prominent risk factor. The results can be used by healthcare services to improve the care and follow-up persons with dementia and their ICGs receive by mapping caregiver burden and aggressive behavior from the person with dementia and by tailoring services to reduce and prevent high burden and aggression. Furthermore, it seems that municipal health services should consider assigning a personal health service contact to people with dementia and their ICGs as a relatively simple measure with the potential to reduce the risk of abuse.

Future research should further explore the dynamics of how risk factors affect elder abuse, using a longitudinal mixed-method approach to confirm causation and ascertain what underlies these associations across countries and cultures. Specifically, such research will provide further evidence of how social context affects elder abuse and interacts with factors in other contexts. There is also a need to initiate and implement individually tailored interventions that include elder abuse as an issue and focus on reducing the subjective caregiver burden and preventing and managing aggression from the person with dementia.

## Figures and Tables

**Figure 1 ijerph-20-02823-f001:**
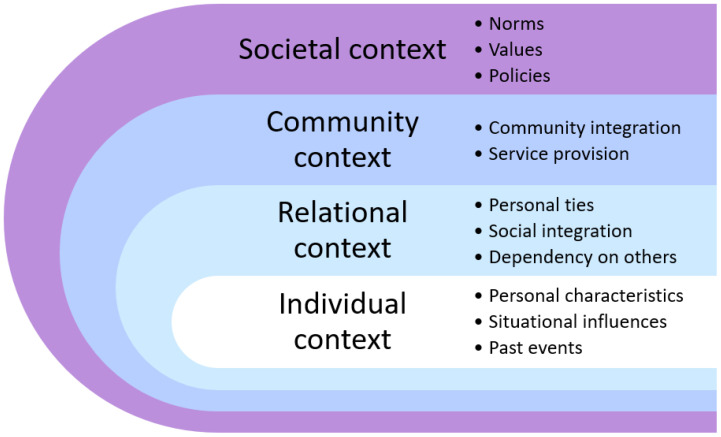
The contextual theory of elder abuse [7].

**Figure 2 ijerph-20-02823-f002:**
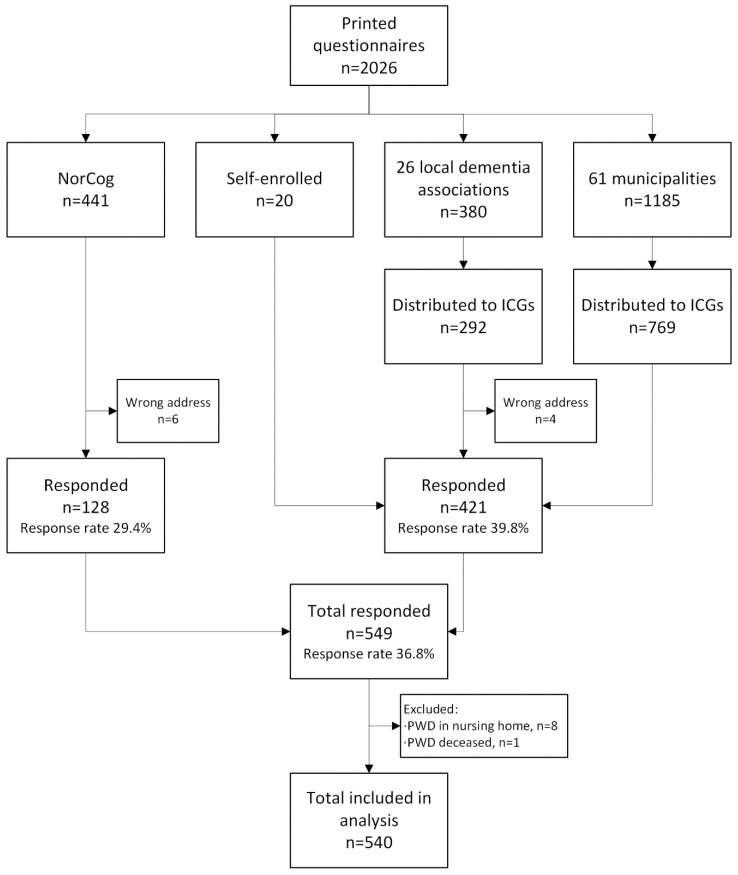
Recruitment. ICG: informal caregiver, PWD: person with dementia, NorCog: the Norwegian Registry of Persons Assessed for Cognitive Symptoms.

**Figure 3 ijerph-20-02823-f003:**
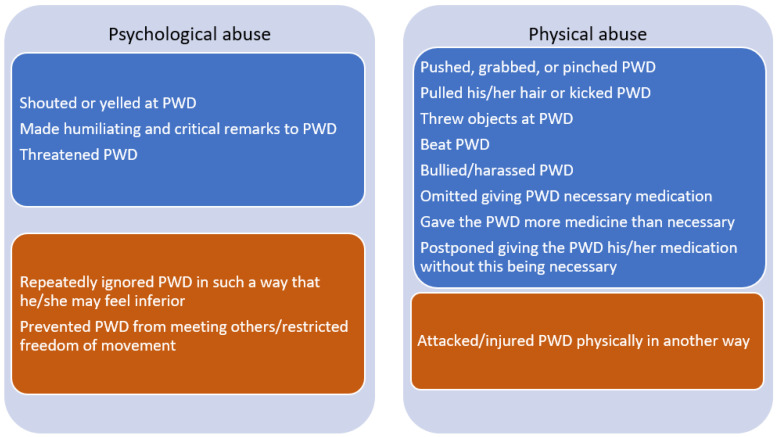
Items measuring abusive episodes. Blue adapted from Botngård et al. [29], orange adapted from Sandmoe et al. [30]. PWD: person with dementia.

**Figure 4 ijerph-20-02823-f004:**
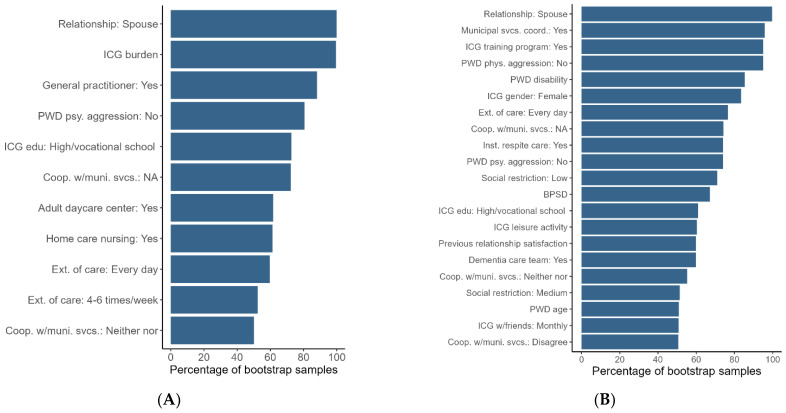
Covariates selected in >50% of the 1000 bootstrap samples in percent not null. Outcome variables: (**A**) psychological abuse; (**B**) physical abuse.

**Figure 5 ijerph-20-02823-f005:**
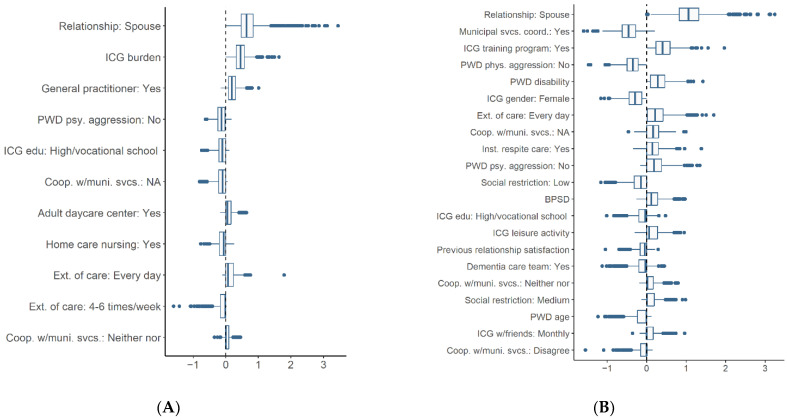
Boxplots of standardized regression coefficients for variables selected by lasso in >50% of 1000 bootstrap samples. Outcome variables: (**A**) psychological abuse; (**B**) physical abuse. Median (horizontal line), 50% middle values (quartile 2–3) (box), extremities (±1.5 interquartile range) (whiskers), outliers (dots).

**Table 1 ijerph-20-02823-t001:** Descriptions of the covariates.

	Variable	Questions/Measure	Categorizations/Score
Individual context, ICG	ICG gender	Your gender?	1. Female; 2. Male
	ICG age	Your birth year?	2021—birth year
	ICG employment	What is your current main activity? (Employment includes studies, not working includes unemployed, retired, disabled or similar)	1. Full-time employment; 2. Working part-time; 3. Not working
	ICG health	All in all, how do you assess your health? Would you say it is: 1. Excellent; 2. Good; 3. Average; 4. Poor; 5. Very poor. Recoded 4–5 = 1, 3 = 2, 1–2 = 3.	1. Poor; 2. Average; 3. Good
	ICG mental health	Measured by the Patient Health Questionnaire for Depression and Anxiety (the PHQ–4) [34]. Four questions. Question scoring from 0 “Not at all” (bothered) to 3 (bothered) “Nearly every day”.	Sum score: 0–12. A higher score indicates a higher risk of anxiety and/or depression.
	ICG education	What is your highest completed level of education?	1. High/vocational school level or lower; 2. University/college level or higher
	ICG economy	How easy or difficult is it for your household to manage financially on a daily basis, given your income?	1. Hard; 2. Easy
Individual context, PWD	PWD gender	What gender is the PWD?	1. Female; 2. Male
	PWD age	What year were they born?	2021—birth year
	Dementia duration	Approximately how long has he/she had symptoms of dementia?	0. ≤2 years; 1. >2–4 years 2. >4 years
	PWD disability	Measured by Rapid Disability Rating Scale 2 (RDRS2) [35] with 18 questions (without “degree of special problems subscale”—q18–20). Questions scored from 1 “None” (completely independent or normal behavior) to 4 “Total” (the person cannot, will not, or may not perform a behavior or has the most severe form of disability/problem).	Sum score 18–72. Higher score indicates more disability.
	BPSD	Neuropsychiatric Inventory–Questionnaire (NPI-Q) [36]. 12 questions measuring whether a symptom has been present in the past month (Yes/No) and the severity of the present symptom (1—mild, 2—moderate, 3—severe).	Sum score 0–36. Higher score indicates a more severe symptom burden.
	PWD alcohol consumption *	Approximately how often has the PWD consumed alcohol during the past 12 months?	1. Never; 2. Monthly or less often; 3. Weekly
Relational context	Relationship	What is your relationship with the PWD?	1. Spouse/cohabitant/partner; 2. Child, sibling, or other
	Caregiving duration	How long have you been helping them because of their illness?	1. 0–2 years; 2. 3–5 years; 3. 6 years or longer
	Extent of care	How often do you help him/her?	1. ≤Once a week; 2. 2–3 times a week; 3. 4–6 times a week; 4. Every day
	Previous relationship satisfaction	Measured by adapting a partnership satisfaction index [37]. Instructions: Think about the relationship you had with him/her before the illness. How much do you agree with the statement (a) We agreed on what is important in life; (b) We often had conflicts (inverted); (c) She/he often criticized me (inverted); (d) She/he understood me when I had problems. Rated on a scale from 0 “strongly disagree” to 5 “strongly agree.”	Sum score 0–20. Higher score indicates higher satisfaction.
	ICG burden	Caregiver subjective burden measured by the Relative Stress Scale (RSS) [38], 15 questions. The Norwegian version of the RSS is tested among ICGs of persons with dementia [39]. Rated from 0 “Never/not at all” to 4 “Always/considerably.”	Sum score 0–60. Higher score indicates a higher burden.
	PWD psychological aggression	Has the PWD done anything similar towards you? (Following questions regarding psychological abuse towards the PWD)	0. No 1. Yes
	PWD physical aggression	Has the PWD done anything similar towards you? (Following questions regarding physical abuse towards the PWD)	0. No 1. Yes
Community context	ICG friends *	How often are you together with good friends? Do not include members of your own family.	1. Weekly; 2. Monthly; 3. Less often
	ICG leisure activity *	Approximately how often do you do the following in your spare time? (a) Exercises or are physically active so that you breathe heavily and/or sweat; (b) Go to the cinema, theater, concerts, and/or art exhibitions; (c) Participate in social activities in a club, association, or organization; (d) Attend worship services or other religious meetings. Score: 0 “Never,” 1 “ Less often,” 2 “Few times a year,” 3 “Monthly, but not weekly,” 4 “Weekly, but not daily,” 5 “Daily.”	Sum score 0–20. Higher score indicates more leisure activity.
	PWD leisure activity *	How is the PWD’s participation in social and cultural activities (think a weekly average for the year)? Do not include services and activities provided by the municipality. (a) Social activities; (b) Cultural activities	0. No activity; 1. Activity
	Social restriction	Measured by The Modified Social Restriction Scale [40]. This scale consists of two questions regarding whether you have anyone else who can take care of the PWD if you get sick or need a break. Score: 1. “Yes, it will be easy to find someone,” 2. “Yes, I can find some, but it will not be that easy,” 3. “No, there is no one else.”	Score categorization: 2–3 = 1 “Low” (quite easy); 4 = 2 “Medium” (possible, but not easy); 5–6 = 3 “Hard” (no one else)
	Coop. w/municipal svcs.	Thinking of the municipality the PWD receives services from, how much do you agree with the statements: (a) The cooperation between me as a caregiver and the municipality works well; (b) I am consulted on questions regarding services and offers for the person with dementia; (c) I get sufficient information from the municipality about the services the person with dementia receives. Scale: 1 (strongly disagree)—5 (strongly agree) or 100 (Not applicable).	Sum score categorization: 1–7, 102–105, 201, 202 = 1 “Disagree”; 8–10, 106, 203 = 2 “Neither nor”; 11–15, 107–110, 204, 205 = 3 “Agree”; 300 = 4 “Not applicable”
	ICG training program	Have you participated/are you participating in a caregiver training program? (This is a course that consists of lectures and theme-based group discussions where the participants exchange experiences and gain knowledge about dementia, communication, coping, legislation, and services).	1. Yes; 2. No
	ICG other training	Have you received/are you receiving other training about dementia?	1. Yes; 2. No
	ICG support group	Do you participate in a peer support group with other caregivers of PWDs?	1. Yes; 2. No
	PWD education prog.	Has the PWD attended, or do they attend a dementia education program? (This is a course for PWDs that provides knowledge about dementia, stimulates self-management, and is a meeting place for exchanging experiences and mutual support); Does the PWD participate in a peer support group? If “Yes” on one or both questions: Yes; if “No” on both: No	1. Yes; 2. No
	Municipal svcs. contact	Have you been assigned a contact person/coordinator from the municipal health services for you and the PWD to contact?	1. Yes; 2. No
	Do you, due to your caregiver role, or the PWD receive any of the following services or offers?		
	Dementia care team	Follow-up from dementia care team/dementia contact	1. Yes; 2. No
	Adult daycare center	Activity offer, adult day care center, or similar	1. Yes; 2. No
	Institutional respite care	Respite care in an institution (nursing home/care home/assisted living facility)	1. Yes; 2. No
	Home care nursing	Home care nursing	1. Yes; 2. No
	PWD support person	Support contact (a person who helps another person to have an active and meaningful leisure time. These services are publicly funded); Friendly visitor volunteer (a volunteer that meets and does different activities with a PWD). If “Yes” on one or both questions: Yes; if “No” on both: No.	1. Yes; 2. No
	General practitioner	General practitioner (consultation/follow-up)	1. Yes; 2. No
	Hospital follow-up	Follow-up in hospital/outpatient clinic	1. Yes; 2. No

* Questions adapted from the NorCog test battery. PWD = person with dementia; ICG = informal caregiver; BPSD = behavioral and psychological symptoms of dementia; svcs. = services.

**Table 2 ijerph-20-02823-t002:** Descriptive statistics for covariates according to elder abuse subtype in frequency (%) or mean ± SD.

		Psychological Abuse		Physical Abuse	
		No Abuse *n* = 429	Abuse *n* = 108	No Abuse *n* = 486	Abuse *n* = 53
Individual context, ICG	ICG gender: Female (vs. male)	283 (66.6)	78 (75.0)	330 (68.8)	32 (62.8)
	ICG age	66.53 ± 12.24	70.81 ± 8.88	66.68 ± 11.99	74.25 ± 6.13
	ICG employment: Full-time	126 (29.7)	18 (17.6)	141 (29.6)	3 (5.9)
	Part-time	43 (10.1)	7 (6.9)	47 (9.9)	3 (5.9)
	Not working	255 (60.1)	77 (75.5)	289 (60.6)	45 (88.2)
	ICG health: Poor	44 (10.4)	14 (13.1)	52 (10.9)	6 (11.3)
	Average	111 (26.2)	36 (33.6)	129 (26.9)	18 (34.0)
	Good	268 (63.4)	57 (53.3)	298 (62.2)	29 (54.7)
	ICG mental health	2.46 ± 2.24	3.98 ± 3.08	2.65 ± 2.44	3.77 ± 2.83
	ICG education: ≤High school (vs. college/univ.)	252 (59.2)	53 (51.0)	279 (58.0)	27 (52.9)
	ICG economy: Hard (vs. easy)	39 (9.3)	8 (7.7)	41 (8.6)	6 (11.8)
Individual context, PWD	PWD gender: Female (vs. male)	224 (52.5)	32 (30.2)	238 (49.2)	19 (37.3)
	PWD age	79.20 ± 7.81	77.72 ± 7.58	79.14 ± 7.92	76.94 ± 5.92
	Dementia duration: ≤2 years	96 (22.4)	19 (17.8)	105 (21.7)	10 (19.2)
	>2–4 years	159 (37.2)	39 (36.5)	183 (37.7)	16 (30.8)
	>4 years	173 (40.4)	49 (45.8)	197 (40.6)	26 (50.0)
	PWD disability	39.18 ± 9.72	42.01 ± 9.93	39.13 ± 9.64	44.32 ± 10.46
	BPSD	8.51 ± 6.34	9.89 ± 5.53	8.49 ± 6.03	10.67 ± 7.17
	PWD alcohol consumption: Never	147 (34.4)	28 (26.7)	159 (32.9)	16 (31.4)
	≤Monthly	156 (36.5)	40 (38.1)	180 (37.3)	18 (35.3)
	Weekly	124 (29.0)	37 (35.2)	144 29.8	17 (33.3)
Relational context	Relationship: Spouse (vs. child or other)	242 (56.5)	94 (89.5)	288 (59.5)	50 (98.0)
	Caregiving duration: 0–2 years	152 (35.6)	29 (27.6)	166 (34.3)	15 (30.0)
	3–5 years	201 (47.1)	58 (55.2)	233 (48.1)	27 (54.0)
	≥6 year	74 (17.3)	18 (17.1)	85 (17.6)	8 (16.0)
	Extent of care: ≤Once a week	37 (8.7)	3 (2.9)	40 (8.3)	0 (0.0)
	2–3 times/week	76 (17.9)	4 (3.8)	80 (16.6)	0 (0.0)
	4–6 times/week	67 (15.8)	3 (2.9)	68 (14.1)	2 (4.0)
	Every day	245 (57.7)	95 (90.5)	294 (61.0)	48 (96.0)
	Previous relationship satisfaction	3.78 ± 0.96	3.56 ± 0.90	3.75 ± 0.96	3.58 ± 0.90
	ICG burden	23.35 ± 11.37	31.72 ± 9.91	24.36 ± 11.49	30.69 ± 11.39
	PWD psychological aggression: Yes	125 (29.8)	54 (50.5)	158 (33.1)	21 (41.2)
	PWD physical aggression: Yes	29 (6.9)	12 (11.5)	29 (6.2)	12 (22.6)
Community context	ICG w/friends: Weekly	163 (38.4)	28 (26.2)	177 (36.8)	15 (28.9)
	Monthly	139 (32.7)	41 (38.3)	160 (33.3)	20 (38.5)
	Less often	123 (28.9)	38 (35.5)	144 (29.9)	17 (32.7)
	ICG leisure activity	6.99 ± 3.11	6.22 ± 3.46	6.84 ± 3.13	6.80 ± 3.76
	PWD leisure activity: Activity (vs. no activity)	337 (79.3)	80 (76.9)	380 (79.5)	38 (73.1)
	Social restriction: Low	186 (44.1)	26 (24.3)	201 (42.1)	11 (20.8)
	Medium	133 (31.5)	39 (36.5)	152 (31.9)	21 (39.6)
	High	103 (24.4)	42 (39.3)	124 (26.0)	21 (39.6)
	Coop. w/municipal svcs.: Disagree	69 (16.2)	14 (13.2)	78 (16.2)	5 (9.4)
	Neither nor	87 (20.4)	29 (27.4)	102 (21.2)	14 (26.4)
	Agree	239 (56.0)	61 (57.6)	272 (56.6)	29 (54.7)
	Not applicable	32 (7.5)	2 (1.9)	29 (6.0)	5 (9.4)
	ICG training program: Yes	175 (40.8)	52 (48.6)	193 (39.8)	34 (64.2)
	ICG other training: Yes	90 (21.0)	22 (20.6)	103 (21.2)	9 (17.7)
	ICG support group: Yes	75 (17.5)	30 (27.8)	90 (18.6)	15 (28.3)
	PWD education prog.: Yes	61 (14.2)	14 (13.0)	66 (13.6)	9 (17.0)
	Municipal svcs. contact: Yes	281 (65.7)	73 (67.6)	325 (67.0)	30 (56.6)
	Dementia care team: Yes	191 (45.1)	50 (47.2)	222 (46.4)	19 (37.3)
	Daycare center etc.: Yes	224 (52.6)	69 63.9)	260 (53.8)	34 (65.4)
	Institutional respite care: Yes	102 (23.8)	39 (36.5)	116 (24.0)	25 (48.1)
	Home care nursing: Yes	238 (56.5)	45 (41.7)	259 (54.3)	24 (46.2)
	Support person: Yes	52 (12.2)	15 (13.9)	58 (12.0)	9 (17.0)
	General practitioner: Yes	336 (78.9)	95 (88.0)	386 (80.1)	45 (86.5)
	Hospital follow-up: Yes	138 (32.2)	35 (33.0)	157 (32.6)	16 (30.2)

PWD = person with dementia; ICG = informal caregiver; BPSD = behavioral and psychological symptoms of dementia; svcs. = services.

**Table 3 ijerph-20-02823-t003:** Penalized odds ratios (OR) from lasso and percent not null from 1000 bootstrap samples (*n* = 439).

		Psychological Abuse		Physical Abuse	
	Variable	OR	Perc. Not 0	OR	Perc. Not 0
Individual context, ICG	ICG gender: Female (vs. male)	*1*	49.5	0.63	83.5
	ICG age	*1*	17.5	*1*	20.9
	ICG employment: Full-time	*1*	19.9	0.87	47.8
	Part-time	*1*	32.8	*1*	16.2
	Not working	*1*	12.8	*1*	38.3
	ICG health: Poor	*1*	27.7	*1*	37.5
	Average	*1*	26.6	*1*	30.3
	Good	*1*	19.5	*1*	27.7
	ICG mental health	*1*	38.3	*1*	33.0
	ICG education: Lower (vs. higher)	*1*	72.7	*1*	61.0
	ICG economy: Hard (vs. easy)	*1*	43.0	*1*	43.4
Individual context, PWD	PWD gender: Female (vs. male)	*1*	30.7	*1*	14.2
	PWD age	*1*	32.0	*1*	50.9
	Dementia duration: ≤2 years	*1*	21.9	1	47.3
	>2–4 years	*1*	24.4	*1*	24.0
	>4 years	*1*	22.0	*1*	30.7
	PWD disability	*1*	37.7	1.03	85.4
	BPSD	*1*	24.7	1.02	67.1
	PWD alcohol consumption: Never	*1*	41.1	*1*	34.1
	≤Monthly	*1*	24.9	*1*	35.7
	Weekly	*1*	28.1	*1*	26.4
Relational context	Relationship: Spouse (vs. child or other)	2.82	100.0	6.45	99.7
	Caregiving duration: 0–2 years	*1*	38.0	*1*	20.8
	3–5 years	*1*	31.8	*1*	36.7
	≥6 year	*1*	24.0	*1*	36.7
	Extent of care: ≤Once a week	*1*	19.9	*1*	17.9
	2–3 times/week	*1*	16.2	*1*	18.7
	4–6 times/week	*1*	52.4	*1*	3.1
	Every day	1.18	59.6	1.46	76.5
	Previous relationship satisfaction	*1*	46.8	0.96	59.9
	ICG burden	1.04	99.5	*1*	27.8
	PWD psychological aggression: No	0.92	80.5	1.22	74.0
	PWD physical aggression: No	*1*	39.5	0.33	95.0
Community context	ICG w/friends: Weekly	*1*	44.7	*1*	39.4
	Monthly	*1*	42.8	1.03	50.8
	Less often	*1*	17.4	*1*	24.1
	ICG leisure activity	*1*	29.4	1	60.3
	PWD leisure activity: No activity (vs. activity)	*1*	48.8	*1*	45.6
	Social restriction: Low	*1*	41.1	0.72	71.0
	Medium	*1*	47.1	*1*	51.4
	High	*1*	19.6	*1*	19.2
	Coop. w/municipal svcs.: Disagree	*1*	23.3	*1*	50.6
	Neither nor	*1*	50.1	1.02	55.2
	Agree	*1*	13.6	*1*	18.6
	Not applicable	*1*	72.3	1.56	74.2
	ICG training program: Yes	*1*	46.3	1.91	95.0
	ICG other training: Yes	*1*	35.3	*1*	40.8
	ICG support group: Yes	*1*	37.4	*1*	43.3
	PWD education prog.: Yes	*1*	45.1	*1*	43.3
	Municipal svcs. contact: Yes	*1*	38.3	0.47	95.8
	Dementia care team: Yes	*1*	37.2	0.97	59.8
	Adult daycare center: Yes	*1*	61.7	*1*	42.8
	Institutional respite care: Yes	*1*	38.6	1.26	74.0
	Home care nursing: Yes	*1*	61.3	*1*	40.5
	Support person: Yes	*1*	35.4	*1*	38.3
	General practitioner: Yes	1.15	88.1	*1*	42.8
	Hospital follow-up: Yes	*1*	34.9	*1*	38.7

OR in italics = variable not selected by lasso. PWD = person with dementia; ICG = informal caregiver; BPSD = behavioral and psychological symptoms of dementia; svcs. = services.

## Data Availability

The data presented in this study might be provided by the corresponding author if granted approval from the Regional Committee for Medical Research Ethics Central Norway. The data are not publicly available due to ethical restrictions. Due to the nature of this research, participants of this study were not asked to agree for their data to be shared publicly.

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
