# Peer review of "Contextual Factors Associated with Abuse of Home-Dwelling Persons with Dementia: A Cross-Sectional Exploratory Study of Informal Caregivers"

_ijerph, 2023, doi:10.3390/ijerph20042823_

Round 1
Reviewer 1 Report
The research presented an important issue of factors associated with the abuse of home-dwelling people with dementia.
Although the article is socially crucial, I have some suggestions, pointed out below:
Abstract
ICG shortcut is not explained.
Introduction
The literature review part should be expanded and focused also on more practical aspects, focused on prevention and interventions.
ICG shortcut is still not explained.
Participants
This part focuses only on inclusion criteria, not on participants. I suggest putting the information about the study group here.
How were the inclusion criteria measured?
What were the exclusion criteria?
Figure 2 – the shortcut “CorCog” should be explained.
Data collection
In this part, the inclusion/exclusion criteria should be mentioned.
Did the participants provide written consent? If yes, please justify how the research was anonymous.
How long did it take to complete the surveys?
This is not clear to me how people with dementia and their caregivers were recruited through Norwegian Registry of Persons Assessed for Cognitive Symptoms (NorCog) and municipal healthcare services. How were the tools distributed from those organizations to respondents?
Measurements
I suggest providing more detailed information about the measurement tools, especially the measurement of abuse. The Authors state “Abuse was measured by five items concerning psychological abuse and nine items concerning physical abuse. The items were adapted from two Norwegian cross-sectional studies of elder abuse conducted in nursing homes [28] and among home-dwelling older persons [29].” I suggest a more detailed description and to provide the information about reliability and validity of this measurement – in general, and in the current study.
Did caregivers fill all the measures indicated in the table 1?
The perception of the data from table 1 is very difficult; I suggest a normal text with the description of every variable and its measurement. Please provide the reliability of the tools from the current study if applicable.
What was the level of missing data in the study? Did you check as they were missing at random? I recommend checking. Please justify the strategy chosen for missing data handling.
I suggest underlining the research questions and/or hypotheses.
Results
Figure 1. Covariates selected in >50% of the 1,000 bootstrap samples in percent not null. Outcome variables: A) Psychological abuse; B) Physical abuse – I suppose this should be a Figure 4.
Discussion
The discussion should be completely revised and enriched.
I suggest carefully rethinking using the word “protective” and “risk” factors. The Authors do not explore the mechanisms of the relationships, then treating gender as a protective or risk factor seems to be too simplistic. There is no evidence of a causal effect in the study. How being a man/woman can be a risk factor for abusing anyone? These conclusions are far from justified.
Those are only apparent relations and the true cause-and-effect relationships and mechanisms remain unknown. The result needs to be explained and discussed carefully.
What are the practical implication of the study? The authors state that “The results can be used by healthcare services to improve the care and follow up persons with dementia and their ICGs receive and to inform the development of interventions to address and prevent elder abuse.” Please describe how the result can be used for such purposes.
Please include some information about existing interventions in the field of abuse od PWD.
Reviewer 2 Report
Dear authors,
Thank you for submitting your journal article - Contextual factors associated with abuse of home-dwelling persons with dementia: A cross-sectional exploratory study of informal caregivers, for consideration.
I found it to be a very well written and conceptualised article, though addressing a few minor issues will improve the article overall. A caveat - I am not able to comment in regards to your research methodology as I am not familiar with penalized logistic regression using lasso.
Firstly, although I can appreciate that it would be difficult to include all forms of elder abuse in your study as it would increase the length of the survey for participants, it would be good to justify why only physical and psychological abuse included, for example, in my country, the two highest forms of abuse reported tend to be psychological and financial abuse.
You provided some good background information about theoretical models of elder abuse, referencing Roberto and Teaster etc. I think you could also consider including some literature and explanation around the bi-focal model of abuse, which highlights both the older person experiencing the abuse and the perpetration, as this would link well with your survey design, where you are looking at risk and protective factors for both of these participant types. Some suggestions include:
Lawrence B. Schiamberg PhD & Daphna Gans MA (1999) An Ecological Framework for Contextual Risk Factors in Elder Abuse by Adult Children, Journal of Elder Abuse & Neglect, 11:1, 79-103, DOI: 10.1300/J084v11n01_05
Von Heydrich L, Schiamberg LB, Chee G. Social-Relational Risk Factors for Predicting Elder Physical Abuse: An Ecological Bi-Focal Model. The International Journal of Aging and Human Development. 2012;75(1):71-94. doi:10.2190/AG.75.1.f
One interesting finding was that the ICG attending a caregier training program was a risk factor. You propose that this might be due to gaining more knowledge and awareness, thus reporting more abuse. This could also be linked to literature around the Dunning-Kruger Effect, which is a cognitive bias where people with low ability, expertise or experience regarding a certain type of task or knowledge area tend to overestimate their ability of knowledge.
- Kruger, J., & Dunning, D. (1999). Unskilled and unaware of it: How difficulties in recognizing one's own incompetence lead to inflated self-assessments. Journal of Personality and Social Psychology, 77(6), 1121-1134. https://doi.org/10.1037/0022-3514.77.6.1121
Another finding, that being followed up with the GP is a risk factor for psychological absuse, is also interesting. Though, the suggestion that this could be due to the person with dementia refusing to see the GP, causing the ICG to feel justified in shouting at or threatening them to make them cooperate is very speculative. I would suggest either removing this point or adding evidence to support it in the form of referencing other studies with similar findings.
Round 2
Reviewer 1 Report
Thank you for improving the manuscript. The majority of my suggestions have been implemented. However, I recommend conducting Little's Missing Completely at Random (MCAR) Test instead of manual dataset checking for missing data.
Author Response
Point 1: Thank you for improving the manuscript. The majority of my suggestions have been implemented. However, I recommend conducting Little's Missing Completely at Random (MCAR) Test instead of manual dataset checking for missing data.
Response 1: Thank you very much for this suggestion. We have carried out Little’s MCAR test for each of the sum score scales and added the following to the manuscript (lines 215-218):
Little’s test for missing completely at random (MCAR) was performed on all items within each sum score scale with no significant results (p ranged from 0.08 to 0.99), indicating that missing items can be treated as MCAR.